# Assessment Potential of Zooplankton to Establish Reference Conditions in Lowland Temperate Lakes

Agnieszka Ochocka [1],* and Maciej Karpowicz [2]

1    Department of Freshwater Protection, Institute of Environmental Protection-National Research Institute, Krucza 5/11D, 00-548 Warsaw, Poland
2    Department of Hydrobiology, Faculty of Biology, University of Białystok, Ciołkowskiego 1J, 15-245 Białystok, Poland; m.karpowicz@uwb.edu.pl
*    Correspondence: a.ochocka@ios.edu.pl

**Abstract:** Zooplankton community data from 45 dimictic lakes, representing homogenous abiotic conditions, were used to distinguish indicator taxa of near-pristine, reference lakes with low anthropopression. Reference conditions were selected based on natural land use in the catchment, lack of or low human activity, and the absence of point sources of pollution, as well as good water quality. According to these criteria, six lakes were designated references and all represent mesotrophic conditions. Reference lakes had a low abundance of Cyclopoida and Rotifera, and significantly lower biomass compared to non-reference lakes. We have found that species characteristic of the reference lake were: *Bosmina* (*Eubosmina*) *coregoni*, *Ascomorpha ecaudis*, *Collotheca pelagica*, and *Gastropus stylifer*. The species responsible for differences among reference and non-reference lakes were *Keratella tecta*, *Pompholyx sulcata*, and *Ascomorpha saltans*, which are considered typical for eutrophic waters.

**Keywords:** zooplankton community; indicator species; reference conditions; ecological status; Water Framework Directive

## 1. Introduction

Zooplankton are a fundamental component of the pelagic trophic webs in lakes due to their pivotal role in energy transfer from primary producers to higher trophic levels [1]. Eutrophication processes cause great changes in zooplankton taxonomic structure, abundance, biomass, and size structure, which may serve as early warning indicators of water quality deterioration [2–4]. In particular, oligotrophic lakes are distinguished from eutrophic waterbodies by their low biomass and high species richness [5]. Gliwicz [6] showed that nutrient enrichment of the environment can shift the size structure of zooplankton communities towards small-bodied species, which had a higher reproduction rate. Zooplankton has been long recognized as a reliable indicator of lake trophic status [2,7–10]. Nowadays, the high indicative value of zooplankton to assess trophic conditions is widely accepted [3,11–15].

The Water Framework Directive (WFD) [16] was adopted in the European Union in 2000, to achieve "good ecological status" in all waters (lakes, rivers, coastal, and transitional waters) by 2015. Ecological status is an expression of the quality of the structure and functioning of aquatic ecosystems, considered as a deviation from the non-impacted reference conditions. Excess nutrient supply has become one of the greatest threats to aquatic ecosystems. Therefore, to achieve the WFD goals, it is necessary to reduce and prevent anthropogenically-induced eutrophication. The assessment of water quality is based on four biological quality elements (BQE), namely, phytoplankton, macrophytes, and phytobentos (sometimes used separately), invertebrates, and fish, as well as the supporting physico-chemical and hydromorphological parameters. The WFD compliance assessment does not involve zooplankton communities as a BQE, which seems to be not justified [17–19]. Although zooplankton communities are widely acknowledged as a good indicator of trophic

status in lakes, they have been scarcely used for ecological status assessment and there are only a few zooplankton-based methods elaborated until recently [20,21].

Trophic and ecological status, although they are usually closely related, are not equivalents. For assessment of ecological status, trophic state per se is not essential but determining whether changes in the ecosystem are of natural background or result from anthropogenic disturbances. Consequently, mesotrophic lakes, if anthropogenically impacted, may have deteriorated ecological status, while eutrophic lakes, if slightly deviated from natural conditions, may represent good ecological conditions. When employing this approach, reference conditions can represent different trophic status, such as a high trophic level. Thus, a crucial step when assessing the ecological status is the establishment of reliable reference conditions, which should be type-specific and reflect natural features of ecosystems. There are a lot of approaches used to establish reference conditions [22,23], which differ among European countries due to the different physical-geographical parameters in catchments (geology, altitude), lake morphometry (e.g., depth, area) [24], and climate features. According to the guidance document of the European Commission, elaborated by Working Group within the Common Implementation Strategy (CIS) [25], these approaches include (i) the spatially based method using data from existing undisturbed or minimally disturbed sites "the best of existing", (ii) predictive modeling using available data within a region, (iii) a temporally based method using historical data, and (iv) expert judgment. Because in some regions of Poland there are still lakes that are considered relatively well-preserved with low pressure and no signs of alteration, we adopted the spatial method to establish the reference conditions.

In this study, we proposed recognizing the potential of zooplankton taxa to reflect reference conditions in lowland temperate lakes of Central Europe. We tested whether zooplankton taxa, which are widely recognized as good indicators of trophic status, can also be used to determine reference conditions in lowland temperate lakes.

## 2. Materials and Methods

### 2.1. Sampling Sites and Data Collection

We analyzed zooplankton, water chemistry, and catchment pressure data from 45 lakes located in north-eastern Poland (see Figure 1). The lakes were selected to represent homogenous abiotic conditions, i.e., lowland, highly alkaline, stratified ecosystems on calcareous deposits [26], as well as the entire gradient of ecological conditions in terms of trophic level and anthropogenic impacts. The analyzed lakes are deep, dimictic with summer stratification, winter inverse stratification, and spring and autumn overturn [27], periodically freezing with an ice cover usually lying from mid-December to mid-March [28]. They are lowland water bodies with a mean depth ranging from 4 to 13 m and a maximum depth ranging from 12 to 57 m.

Samples were collected during the summer stagnation period in 2012–2015. Seven lakes were investigated three times, 27 lakes were investigated two times, and 11 lakes were investigated once during this period. Ultimately, the dataset included 86 lake-years from 45 lakes, including repeated sampling. The sampling sites were located close to the deepest part of each lake. The samples for chemical and zooplankton analysis were collected at intervals of 1 m depth from the surface to the bottom of the epilimnion layer using a 2.6-L Limnos sampler. The volume of filtered water depended on the epilimnion thickness of the lake and ranged between 10 to 21 L. Water was filtered through a plankton net with a mesh size of 30 μm and fixed with Lugol's solution and 4% of formaldehyde.

During our field surveys, we determined water transparency by Secchi disc visibility (SD). Total phosphorus (TP) and total nitrogen (TN) concentrations were determined in the laboratory using standard methods [29]. The concentration of chlorophyll-*a* (Chl-*a*) was determined using the spectrophotometric method [30]. The trophic state index was based on TP, SD, Chl-*a* and SD according to the formulas: TSITP = $14.43 \times \ln(TP) + 4.15$; TSISD = $60 - 14.43 \times \ln(SD)$, TSIChl-*a* = $9.81 \times \ln(Chl\text{-}a) + 30.6$ [31]; TSITN = $54.45 + 14.43 \times \ln(TN)$ [31,32]. The trophic state index (TSI) was calculated as the average of the four above indices using the procedure

described by Kratzer and Brezonik [32]. Lakes with a TSI between 40–50 were classified as mesotrophic, and those with TSI above 50 were regarded as eutrophic.

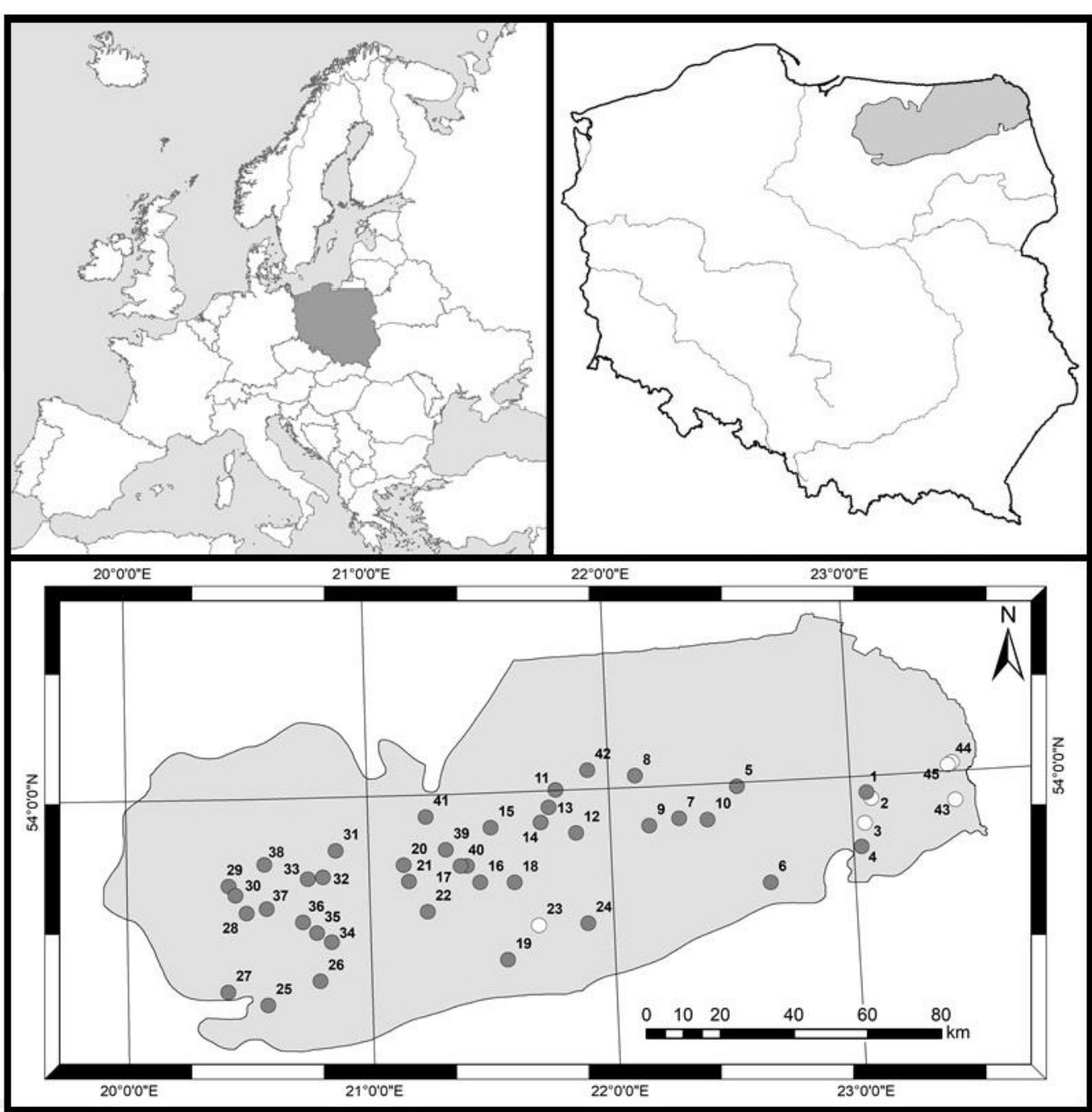

**Figure 1.** Location of studied lakes in East Baltic Lake District. The gray line shows the largest rivers in Poland. The numbers refer to the lake names: 1—Blizno; 2—Busznica, 3—Kalejty, 4—Sajno; 5—Olecko Małe; 6—Rajgrodzkie; 7—Łaśmiady; 8—Gawlik; 9—Garbaś; 10—Zdrężno; 11—Niegocin; 12—Buwełno; 13—Boczne; 14—Jagodne; 15—Ryńskie; 16—Majcz Wielki; 17—Kuc; 18—Mikołajskie; 19—Nidzkie; 20—Lampackie; 21—Piłakno; 22—Gant; 23—Jegocin; 24—Roś; 25—Omulew; 26—Świętajno; 27—Maróz; 28—Bartąg; 29—Ukiel; 30—Kortowskie; 31—Dadaj; 32—Tumiańskie; 33—Kierźlińskie; 34—Leleskie; 35—Kalwa; 36—Purdy; 37—Linowskie; 38—Wadąg; 39—Czos; 40—Probarskie; 41—Kiersztanowskie; 42—Kruklin; 43—Brożane; 44—Wiłkokuk; 45—Zelwa. Reference lakes are marked with white circles, while non reference lakes are highlighted with grey circle.

Crustaceans and rotifers were identified to species. The length of at least 15 individuals was measured for each species. We used the wet weight-length relationships for crustaceans to estimate their biomass by applying the equation proposed by Balushkina and Vinberg [33]. Rotifer biomass was determined following the equation suggested by Ejsmont-Karabin [34].

### 2.2. Selection of Reference Lakes

When selecting lakes for sampling, those that met the criteria for reference lakes were included. The selection of the reference lakes was based on the spatial approach—"the best of existing" using information on land use, human pressure, and water quality. This approach is recommended by CIS, [25] supporting WFD implementation and has been used in the previous studies [35–37]. The criteria for reference lakes included (i) natural land use in the catchment (>80% area of forests or wetlands, lack of villages in direct contact with the shoreline, no urban areas), (ii) no point sources of pollution in the total catchment, (iii) lack of or no intensive recreational use, and (iv) high/good water quality according to official water quality classification.

The data on water quality indicators, which were used to select the reference lakes, were collected from the State Environmental Monitoring program; these data were obtained in 2009–2012. The land use in the catchments of the analyzed lakes was defined based on the CORINE Land Cover 2018 [38].

Based on pressure criteria and physicochemical water parameters, six lakes (17 lake-years) were designated as references. The areas of the total catchment of these lakes were almost entirely forested—the natural land use ranging from 91 to 100% of the catchment (Table 1). There were no point resources of pollution nor urban areas in the catchments, while agricultural areas and pastures occupied no more than 1.5% of the catchment area.

**Table 1.** Mean values of physicochemical and biological variables, and catchment land use parameters between reference and non-reference lakes. Minimum and maximum values are given in parenthesis. One-way ANOVA and Fisher's F test were used to determine differences between tested lakes.

| Parameters | Reference | Non-Reference | One-Way ANOVA Results | |
|---|---|---|---|---|
| | n = 17 Lake-Years | n = 69 Lake-Years | F | p |
| TP [mg L$^{-1}$] | 0.022 (0.010–0.054) | 0.043 (0.012–0.093) | 17.4 | <0.001 |
| TN [mg L$^{-1}$] | 0.64 (0.15–0.94) | 0.92 (0.41–1.65) | 18.5 | <0.001 |
| Chl-*a* [µg L$^{-1}$] | 3.4 (1.2–6.2) | 18.5 (1.3–64.2) | 13.3 | 0.001 |
| SD [m] | 4.6 (2.0–7.0) | 2.1 (0.5–7.0) | 51.3 | <0.001 |
| Natural land use (forests, wetlands and other waterbodies) | 96.6 | 47.7 | 119.2 | <0.001 |
| Continuous urban fabric | 0 | 0.5 | 4.5 | 0.037 |
| Agricultural areas and discontinous urban fabric | 1.5 | 40.8 | 89.4 | <0.001 |
| Pastures | 1.5 | 5.7 | 18.0 | <0.001 |
| Land principally occupied by agriculture with significant areas of natural vegetation | 0.4 | 5.3 | 19.0 | <0.001 |

### 2.3. Statistical Methods

Canonical community ordination techniques [39] were used to establish main gradients in zooplankton and environmental data. The Detrended Correspondence Analysis (DCA) was performed to recognize the type of the data distribution (linear vs. unimodal). Due to a short gradient of biological data, the Redundancy Analysis RDA [39] was used to determine the relationships between environmental variables and zooplankton taxonomic composition in the reference lakes and lakes representing the whole spectrum of water quality (non-reference). This analysis was performed using the abundance of zooplankton dominant species (>5% of the total number of zooplankton individuals). The significance of environmental variables related to the biota was tested with the Monte Carlo permutation with automatic selection and permutation under the full model. The RDA analysis was performed using CANOCO 4.5 [39].

The similarity percentage analysis (SIMPER) was performed to identify species that most contributed to the differences between the two analyzed groups of lakes (reference versus non-reference). The SIMPER analysis was performed using PAST software [40]. Species discriminating between reference and non-reference lakes were identified using the indicator value (IndVal), which is based on the species' relative abundance compared to the relative frequency of occurrence of the species in each group. The IndVal values range between 0 and 1. Species with values $\geq$ 0.5 and significance ($p < 0.05$) are considered Indicators [41,42]. We calculated IndVal values using the labdsr package [43] R 4.2.1 version [44].

Differences between reference and non-reference lakes based on physicochemical, biological variables, and catchment land use parameters were determined using one-way ANOVA with Fisher's F test. We applied the Mann–Whitney U test to compare the distribution of the abundance of zooplankton groups (Cladocera, Calanoida, Cyclopoida, and Rotifera) and the biomass of Crustacea and Rotifera between reference and non-reference lakes.

## 3. Results

### 3.1. Physicochemical Characteristics and Trophic Variables

The physicochemical parameters significantly differed between the reference lakes and non-reference lakes (Table 1). Total phosphorus, total nitrogen, chlorophyll-*a*, and Secchi disc visibility, were lower in reference than in non-reference lakes (Table 1). Total phosphorus in reference lakes ranged from 0.01 to 0.054 mg L$^{-1}$, while TN did not exceed 0.94 mg L$^{-1}$. In reference lakes, the chlorophyll-*a* concentration varied in the range 1.2–6.2 µg L$^{-1}$ and the water transparency ranging from 2 to 7 m.

The trophic state index (TSI) indicated that the reference lakes were mesotrophic, while among non-reference lakes, 12 lake-years were classified as mesotrophic and 57 lake-years were eutrophic (Table S1).

### 3.2. Zooplankton Assemblage Characteristic

In 45 lakes, we identified 81 zooplankton species (Table S2), including 18 species of Cladocera, 5 species of Calanoida, 8 species of Cyclopoida, and 50 species of Rotifera. The abundance of Cyclopoida was significantly lower in the reference lakes (Mann–Whitney U-test, Z = 4.79; $p < 0.001$; Figure 2b). The abundance of Cyclopoida in the non-reference lakes was 134.4 $\pm$ 103.4 ind. L$^{-1}$, while in the reference lakes it accounted for 42.9 $\pm$ 19.7 ind. L$^{-1}$. In both groups of lakes, the most frequently occurring species were *Thermocyclops oithonoides* and *Mesocyclops leuckarti*. No significant differences between the reference and non-reference lakes were found in the abundance of Cladocera and Calanoida (Figure 2a,c). Similarly, the abundance of Rotifera was significantly lower in the reference lakes (Z = 3.79; $p < 0.001$; Figure 2d). The abundance of Rotifera in the non-reference lakes was 664.3 $\pm$ 841.7 ind. L$^{-1}$, while in the reference lakes it constituted 187.9 $\pm$ 79.5 ind. L$^{-1}$.

The mean total biomass of planktonic Crustacea and Rotifera was lower in the reference lakes than in the non-reference lakes (Z = 2.45; $p < 0.05$; Z = 2.77; $p < 0.01$, respectively; Figure 3a,b).

The relationships between environmental variables (TP, TN, Chl-*a*, and SD) and dominant planktonic Crustacea species analyzed using RDA showed that the first two ordination axes explained the 95.5% of variance (Figure 4). The first ordination axis of RDA presented a strong eutrophication gradient, explaining 87.2% of the variation in the Crustacea community attributed to the environmental variables. The reference lakes were grouped on the left side of the ordination graph, and were strongly associated with Secchi disc visibility, while the most non-reference lakes were shifted towards the right side of the graph indicating higher nutrient enrichment and higher chl-a concentrations. Nevertheless, some non-reference lakes were located close to the central part of the graph and approximated reference lakes. All the reference lakes were mesotrophic, while the non-reference lakes were mostly eutrophic, except for 12 lakes that were assessed as mesotrophic (Table S1).

Crustacean taxa associated with the reference lakes included *Bosmina* (*Eubosmina*) *coregoni*, *Eudiaptomus gracilis*, *Calanoida nauplii*, *Heterocope appendiculata*, and *Daphnia cristata*, whereas in lakes rich in nutrients and chlorophyll-*a* associated taxa were *Chydorus sphaericus*, *Thermocyclops oithonoides*, and juvenile forms of Cyclopoida (copepodites).

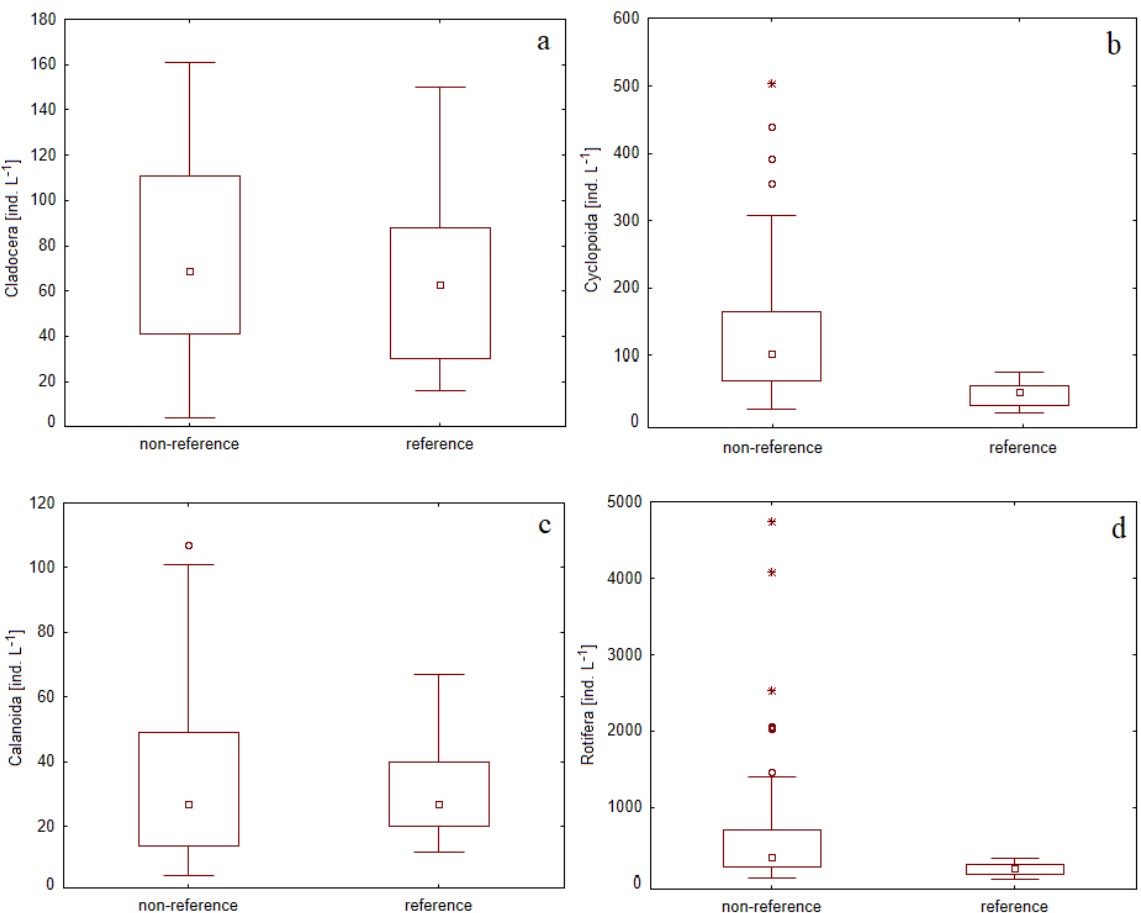

**Figure 2.** Distribution of (**a**) Cladocera, (**b**) Cyclopoida, (**c**) Calanoida, (**d**) Rotifera abundance (ind. L$^{-1}$) in reference (n = 17) and non-reference (n = 69) lakes. Boxplots present 25–75th percentiles with median (squares), whiskers are range, circles are outliers, and asterisks are extreme values.

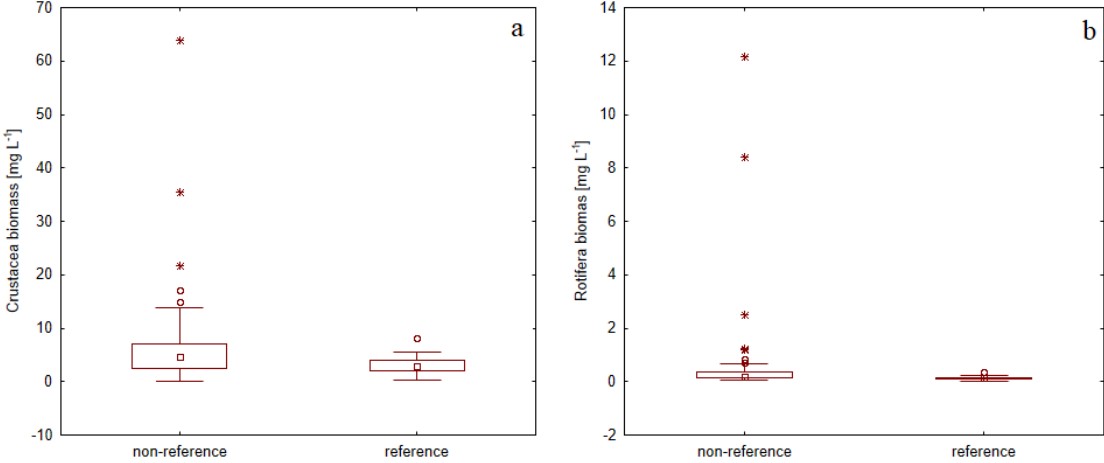

**Figure 3.** Biomass distribution of Crustacea (**a**) and Rotifera (**b**) in the reference (n = 17) and non-reference (n = 69) lakes. Boxplots present 25–75th percentiles with median (squares), whiskers are range, and circles are outliers and asterisks are extreme values.

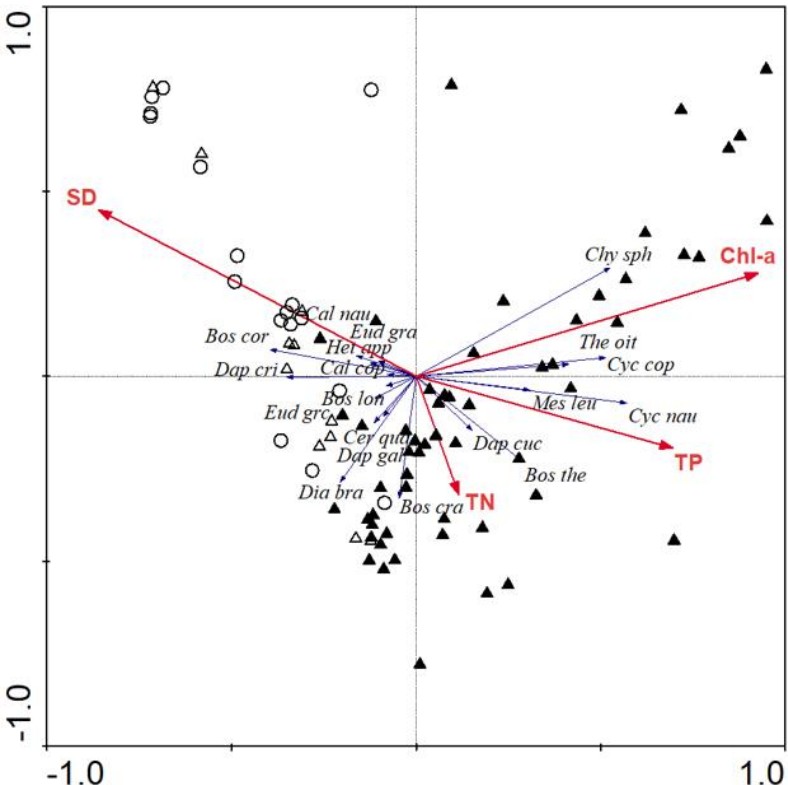

**Figure 4.** The distribution of the reference (circles) and non-reference lakes (triangles) and the Crustacea species (blue arrays) in the RDA space determined by four environmental variables (red arrays): Secchi disc visibility (SD), chlorophyll-*a* concentration (Chl–*a*), total phosphorus (TP), total nitrogen (TN) and abundance of Crustacea species: Bos cor—*Bosmina* (*Eubosmina*) *coregoni*; Bos the—*Bosmina* (*Eubosmina*) *coregoni* var. *thersites*; Bos cra—*Bosmina* (*Eubosmina*) *crassicornis*; Bos lon—*Bosmina* (*Bosmina*) *longirostris*; Cal cop—Calanoida copepodits; Cal nau—Calanoida nauplii; Cer qua—*Ceriodaphnia quadrangula*; Chy sph—*Chydorus sphaericus*; Cyc cop—Cyclopoida copepodits; Cyc nau—Cyclopoida nauplii; Dap cri—*Daphnia cristata*; Dap gal—*Daphnia galeata*; Dia bra—*Diaphanosoma brachyurum*; Eud gra—*Eudiaptomus gracilis*; Eud grc—*Eudiaptomus graciloides*; Het app—*Heterocope appendiculata*; Mes leu—*Mesocyclops leuckarti*; The oit—*Thermocyclops oithonoides*. Mesotrophic lakes are marked in white, while eutrophic lakes are highlighted in black.

Concerning analogous analysis for Rotifera species, the first two ordination axes explained 99% of the relationship between the environmental variables and Rotifera species. (Figure 5). The first axis explained 96.8% of the variation in Rotifera communities related to the environmental variables. The reference lakes were grouped on the left side of the ordination graph and were strongly associated with Secchi disc visibility, while the most non-reference lakes were shifted towards the right side of the graph, indicating higher nutrient enrichment and higher chl-*a* concentrations. Some non-reference lakes were located close to the central part of the graph, i.e., near the reference lakes. Among the Rotifera taxa related to the reference lakes were *Ascomorpha ecaudis*, *Conochilus hippocrepis*, *Polyarthra major*, *Gastropus stylifer*, and *Collotheca pelagica*. *Trichocerca pusilla*, *Trichocerca capucina*, *Keratella quadrata*, *Pompholyx sulcata*, *Brachionus angularis*, *Synchaeta kitina*, and *Brachionus diversicornis* were associated with non-reference lakes.

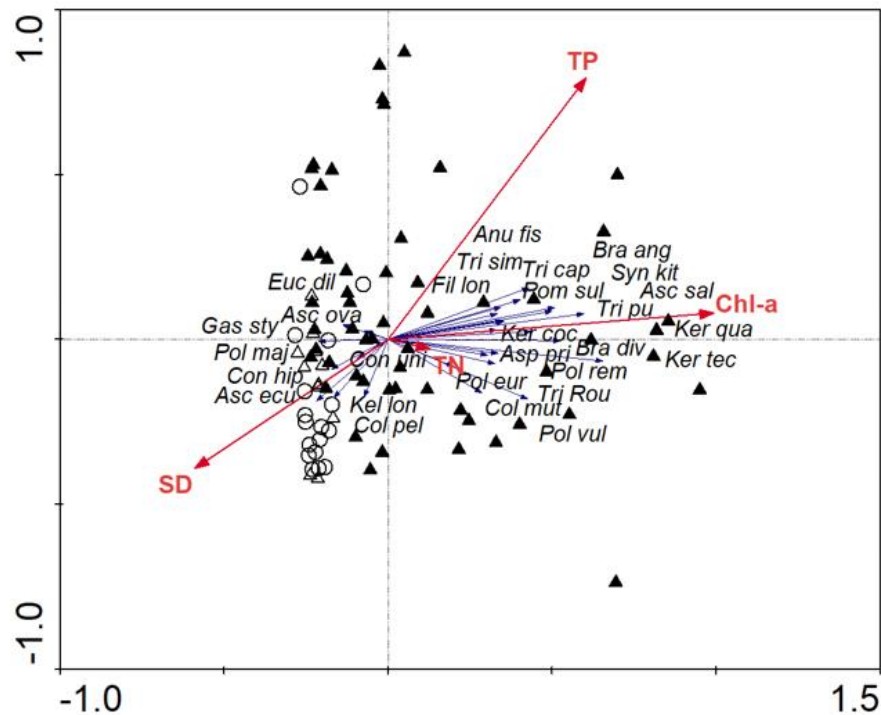

**Figure 5.** The distribution of the reference (circle) and non-reference lakes (triangles) and the Rotifera species (blue arrays) in the RDA space determined by four environmental viariables (red arrays): Secchi disc visibility (SD), chlorophyll-*a* concentration (Chl–*a*), total phosphorus (TP). Rotifera species: Euc dil—*Euchlanis dilatata*; Asc ova—*Ascomorpha ovalis*; Gas sty—*Gastropus stylifer*; Pol maj—*Polyarthra major*; Con uni—*Conochilus unicornis*; Con hip—*Conochilus hippocrepis*; Asc eca—*Ascomorpha ecaudis*; Col pel—*Collotheca pelagica*; Kel lon—*Kellicottia longispina*; Pol vul—*Polyarthra vulgaris*; Col mut—*Collotheca mutabilis*; Pol eur—*Polyarthra euryptera*; Tri Rou—*Trichocerca Rousseleti*; Pol rem—*Polyarthra remata*; Ker tec—*Keratella tecta*; Asp pri—*Asplanchna priodonta*; Bra div—*Brachionus diversicornis*; Ker qua—*Keratella quadrata*; Ker coc—*Keratella cochlearis*; Tri cap—*Trichocerca capucina*; Pom sul—*Pompholyx sulcata*; Fil lon—*Filinia longiseta*; Syn kit—*Synchaeta kitina*; Asc sal—*Ascomorpha saltans*; Tri pus—*Trichocerca pusilla*; Bra ang—*Brachionus angularis*; Tri sim—*Trichocerca similis*; Anu fis—*Anuraeopsis fissa*. Mesotrophic lakes are marked in white, while eutrophic lakes are highlighted in black.

The SIMPER analysis showed that the species *Keratella tecta*, *Pompholyx sulcata*, *Collotheca pelagica*, *Ascomorpha ecaudis*, and *Ascomorpha saltans* were responsible for major variance between reference and non- reference lakes (Table 2).

**Table 2.** Results of the SIMPER analysis for five zooplankton species with highest contribution to differentiation between the reference and non-reference lakes. Data from all samples (lake-years) were analyzed (n = 86).

| Reference Versus Non-Reference Lakes | | |
| --- | --- | --- |
| **Species** | **Contrib.%** | **Cumulative %** |
| *Keratella tecta* | 4.96 | 4.96 |
| *Pompholyx sulcata* | 4.67 | 9.63 |
| *Collotheca pelagica* | 4.57 | 14.20 |
| *Ascomorpha ecaudis* | 3.91 | 18.11 |
| *Ascomorpha saltans* | 3.76 | 21.87 |

The Indicator Value analysis (IndVal; Table 3) using the data on 81 zooplankton taxa revealed four species with significant values ($p < 0.05$) associated with reference conditions,

namely *Bosmina* (*Eubosmina*) *coregoni*, *Ascomorpha ecaudis*, *Collotheca pelagica*, and *Gastropus stylifer*. The rest of the dominant species were related to the non-reference lakes (Table 3).

**Table 3.** Indicator Value (IndVal) of Crustacea and Rotifera species of the reference vs. non-reference conditions. Data from all samples (lake-years) were analyzed (n = 86).

| Species | IndVal | *p* Value | Status |
| --- | --- | --- | --- |
| *Pompholyx sulcata* | 0.854 | 0.001 | Non-reference |
| *Keratella tecta* | 0.725 | 0.001 | Non-reference |
| *Mesocyclops leuckarti* | 0.693 | 0.003 | Non-reference |
| *Thermocyclops oithonoides* | 0.688 | 0.002 | Non-reference |
| *Daphnia cucullata* | 0.628 | 0.014 | Non-reference |
| *Ascomorpha saltans* | 0.591 | 0.009 | Non-reference |
| *Trichocerca pusilla* | 0.463 | 0.007 | Non-reference |
| *Synchaeta kitina* | 0.439 | 0.036 | Non-reference |
| *Keratella quadrata* | 0.417 | 0.016 | Non-reference |
| *Bosmina* (*Eubosmina*) *coregoni var. thersites* | 0.362 | 0.023 | Non-reference |
| *Bosmina* (*Eubosmina*) *coregoni* | 0.723 | 0.001 | Reference |
| *Ascomorpha ecaudis* | 0.634 | 0.001 | Reference |
| *Collotheca pelagica* | 0.583 | 0.001 | Reference |
| *Gastropus stylifer* | 0.564 | 0.021 | Reference |

## 4. Discussion

Lakes with low anthropogenic pressure in the catchments identified in this study as a reference, demonstrated statistically significantly lower trophic conditions than those exposed to anthropogenic influences (non-reference). However, ecosystems with good water quality were also found among the non-reference lakes. Thus, some non-reference lakes could have high or good ecological conditions and similar zooplankton indices as those in the reference lakes. Based on our results, we suggested that reference conditions correspond to mesotrophic status, which is reflected in the characteristics of the zooplankton community inhabiting them.

The clearest difference between zooplankton communities in the reference and non-reference lakes was observed in the abundance of Cyclopoida, which was significantly lower in the reference than in the non-reference lakes. Cyclopoida species are omnivorous predators that are adapted to eutrophic waters [45]. Cyclopoida can win a competition for resources with other crustaceans due to their high tolerance to deterioration of environmental conditions (algae toxins, low dissolved oxygen) [46]. Large-bodied Cyclopoida species get an advantage in lakes with low water transparency because they are less visible to fish predators [45]. In support, Kerfoot and DeMott [47] evidenced that cyclopoid copepods, Rotifera, and small Cladocera are highly abundant in lakes where fish stock is high. The abundance of Rotifera was significantly lower in the reference than in non-reference lakes. Due to their small body size, this group of aquatic animals can successfully avoid predation by planktivorous fish in the pelagic zone. Walz et al. [48] showed that the species composition of Rotifers noticeably responded to the changes in the trophic status of lakes due to the significant correlation of their species occurrence with organic detritus, dead bacteria, and algae. Numerous other studies demonstrated highly positive correlation of Rotifera abundance with tropic level [10,49–51].

We did not find any significant differences in Cladocera and Calanoida abundances between the two groups of lakes. In the pelagic zones of eutrophic lakes, small-bodied cladoceran species such as *Bosmina* spp. and *Chydorus sphaericus* are commonly more abundant than large-bodied *Daphnia* spp. [47]. The replacement of large-bodied Cladocera by small species is a crucial phenomenon in the eutrophication process [1]. The lack of significant differences in Cladocera and Calanoida abundances between the reference and non-reference lakes in our study may be associated with sampling methods. Specifically, we took samples from the epilimnion layer, whereas large-bodied Crustacea (Cladocera and Calanoida) migrate to the deeper water layers during the day [52–54]. For large-

bodied zooplankton, diel vertical migration is a typical strategy that allows them to avoid predation risk and find refuge in the deeper waters during the daytime [55]. Planktivorous fish effectively consume large zooplankton, thus causing the shift of the zooplankton structure towards Rotifera and small Cladocera [56].

Statistical analysis established species associated with the reference and non-reference lakes. The RDA analysis for the Crustacea species indicated that species including *Bosmina* (*Eubosmina*) *coregoni*, *Daphnia cristata*, *Eudiaptomus gracilis*, *Heterocope appendiculata*, and juvenile forms of Calanoida were related to the reference lakes (Figure 4). These species were previously listed by Ejsmont-Karabin and Karabin [12] as indicators of low trophic conditions. Besides, this analysis indicated that Rotifer species including *Ascomorpha ecaudis*, *Conochilus hippocrepis*, *Polyarthra major*, *Gastropus stylifer*, and *Collotheca pelagica* were related to the reference conditions (Figure 5). *Bosmina* (*Eubosmina*) *coregoni* with the highest IndVal (0.723) was associated with low trophic conditions, which is in agreement with Ejsmont-Karabin and Karabin [12]. In addition, two Rotifera species, *Ascomorpha ecaudis* and *Gastropus stylifer*, which Maemets [57] and Ejsmont-Karabin [11] listed as indicators of oligo and mesotrophic lakes, reached a significant IndVal (0.634 and 0.564, respectively) for the reference lakes. *Collotheca pelagica*, with a high and significant IndVal of 0.583, is clearly related to the reference conditions, although it has not been listed as an indicatory species before. However, Karpowicz and Ejsmont-Karabin [58] showed that the abundance of this species was higher in the low trophic conditions. The contribution of *Collotheca pelagica* and *Ascomorpha ecaudis* was significant in the differentiation between the reference and non-reference lakes based on SIMPER analysis. The group of Crustacea species that were related to the non-reference lakes in RDA analysis included *Mesocyclops leuckarti*, *Thermocyclops oithonoides*, *Chydorus sphaericus*, and juvenile forms of Cyclopoida. Before, these taxa were proposed to be indicators of eutrophic conditions by Ejsmont-Karabin and Karabin [12]. Surprisingly, *D. cucullata* was related to the non-reference lakes. This species is one of the most common species in the pelagic zone and occurs in different trophic conditions, from oligotrophy to hypertrophy [59]. Peljer [60] and Gulati [9] showed that *D. cucullata* can be numerous in highly trophic conditions. However, this species migrates to deep waters during the day to avoid planktivorous fish predation. Karpowicz et al. [52] confirmed that larger zooplankton individuals migrate to deep waters, while the smaller individuals remain in the upper layers. In low trophic lakes, where the water is more transparent, *D. cucullata* is more vulnerable to fish predation than in eutrophic lakes, which may be the cause of their greater abundance in the epilimnion of the non-reference lakes. *Mesocyclops leuckarti*, *Thermocyclops oithonoides*, and *Bosmina* (*Eubosmina*) *coregoni thersites* reported by Ejsmont-Karabin and Karabin [12] as indicators of eutrophic waters were also associated with the non-reference lakes.

In the Rotifera community, *Pompholyx sulcata* was associated with non-reference conditions with the highest IndVal. This species is typical of eutrophic lakes [57,61]. The second species with a high IndVal was *Keratella tecta*, which many authors [5,9,11,15,61] regarded as indicator of eutrophic conditions. Ejsmont-Karabin and Karabin [12] indicated two more Rotifera species that are also typical of eutrophic waters, specifically, *Trichocerca pusilla* and *Keratella quadrata*. They were identified as indicators for non-reference lakes based on IndVal results.

SIMPER analysis confirmed that *Keratella tecta*, *Pompholyx sulcata*, and *Ascomorpha saltans* were responsible for differences between the reference and non-reference lakes.

Our research identified species associated with the reference and non-reference lakes. We suggest that the appearance of species associated with non-reference lakes (mostly eutrophic) may be an early warning signal of deterioration of the ecosystem state. The results of our research provide one more piece of evidence that zooplankton can be a reliable and valuable indicator of ecological status and should be included as one of the BQE for the bioassessments.

## 5. Conclusions

The presented results demonstrate that zooplankton taxonomic composition has a high potential to be used as an indicator of reference lakes. Crustacea and Rotifera species, which can be used as indicators for reference conditions, are associated with near-pristine lakes with low anthropopression including *Bosmina* (*Eubosmina*) *coregoni*, *Ascomorpha ecaudis*, *Collotheca pelagica*, and *Gastropus stylifer*, and these species can be used as indicators for reference conditions. Species that are typical of lakes with high trophic status mostly correspond to those with high anthropogenic pressure and can be used as a reliable indicator of water quality deterioration. Therefore, we strongly recommend that the European Union include zooplankton as a BQE in lake monitoring systems.

**Supplementary Materials:** The following are available online at https://www.mdpi.com/article/10.3390/d14060501/s1, Table S1: Trophic status of the studied lakes, Table S2: The list of zooplankton taxa occurred in the analyzed lakes.

**Author Contributions:** A.O.—writing—original draft preparation, conceptualization, field analysis and sampling, analyzed the zooplankton, performed statistical analysis, visualization; M.K.—writing—original draft preparation. All authors have read and agreed to the published version of the manuscript.

**Funding:** This research was supported by the Polish National Science Centre by grant number 2012/07/N/NZ9/01396 and by the grant of the Ministry of Science and Higher Education for the statutory activity of the Institute of Environmental Protection–National Research Institute.

**Institutional Review Board Statement:** Not applicable.

**Data Availability Statement:** The data that support the findings of this study are available from the corresponding author upon reasonable request.

**Acknowledgments:** The Chief Inspectorate for Environmental Protection in Poland is kindly acknowledged as the provider of the physico-chemical monitoring data used in this study to establish the reference lakes. Special thanks go to my colleagues Agnieszka Pasztaleniec, Sebastian Kutyła and Aleksandra Bielczyńska for their help in the collection of samples and Agnieszka Kolada for valuable comments on the manuscript. We extend our thanks to Irina Feniova for the assistance in language editing.

**Conflicts of Interest:** The authors declare no conflict of interest.

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
