# Peer review of "Assessment Potential of Zooplankton to Establish Reference Conditions in Lowland Temperate Lakes"

_diversity, doi:10.3390/d14060501_

Round 1
Reviewer 1 Report
Dear Authors,
This paper is a very important piece of zooplankton researches. Also it was good to see that WFD has to content zooplankton such as US Environmental Protection Agency standards. Please focus on the result all results have to be similar evaluation. I attached my review

Reviewer 2 Report
The concept of reference conditions applied by authors is far from accurate. One could agree in that land use is, indeed, a component to be considered, but "lack or low human activity, and absence of point sources of pollution as well as good water quality" are not. Of course, a reference lake should be as pristine as possible, but it does not mean that those variables are considered to establish reference conditions. In the final part of the 'Introduction' section, authors mention the key question, i.e. "There are a lot of approaches have been used to establish reference conditions which differs among European countries due to the different physical-geographical parameters in catchments (geology, altitude), lake morphometry (e.g. depth, area) and climate features. Therefore, reference conditions should reflect ecological integrity." The process to establish reference conditions is, in any case, well known; functional (i.e., ideal, pristine) profiles of lakes must be established based on the key processes controlling every functional profile. Whether it is possible to find a good reference lake or not depends on its ecological integrity, but the later is not a trait to establish reference conditions. In the end, if authors want to propose zooplankton to serve for establishing reference conditions, one could agree, whenever zooplankton assemblages which are proposed correspond to ideal/pristine reference conditions, and not to impacted situations. We agree in that trophic state and ecological integrity are not equivalent, since there are natural unimpacted lakes that are eutrophic, but trophic state must be also referred to the ideal/pristine reference conditions.
Even if introduction is deeply revised to take this into account, methods cannot without a new research. As a result, it would be necessary to rephrase the title of the work, the entire introduction, the objectives, and the methods, leaving the question about reference conditions as a secondary question throughout the manuscript, and focusing on characterizing trends in a very interesting set of Poland lakes, what is by itself of international interest.
Another example of the need to let reference conditions aside is that authors do not even give information about the straification pattern of lakes, i.e., are all of them monomictic?, is there any lake that develops winter stratification?, etc. Any way, this information should be given, if available even for just a few lakes.
In statistical methods, it should be explicited which analysis are done on which set of lakes. It seems that authors run separate analysis for reference and non-reference lakes, but it is not clear. According to the overall rephrasing that has been proposed, this distinction may be maintained, not to reject the manuscript, since the results are still of international interest, but groups of lakes must be named in a different way (e.g., paying explicit attention to the criteria used to distinguished among them).
'Results', 'Discussion' and 'Conclusions' section should be also revised acoordingly. In particular, discussion and conclusion need a different approach. It is not surprising that no significant differences among lakes have been found, and that some lakes with good ecological status were considered "non-reference" ones. The fact is what authors did (and it is still worthy to be published) was to analyse zooplanckton assemblages and some limnological features in two sets of lakes, more and less disturbed in terms of land use and a couple of criteria more.
Discussion can indeed include some considerations about the original focus of the manuscript (reference conditions), but not as a primary conclusion of the work, since the experimental design is not adequate for that purpose.
Minor English corrections should be done, since a few expressions are not correct.
Round 2
Reviewer 1 Report
Next time please collect samples from the shorline or the soft sediment. It will gives you higher Cladocera taxa number
Author Response
Thank you very much for this suggestion. During next field studies I will also take samples from the shoreline.
Reviewer 2 Report
I acknowledge the effort of the authors to clarify many questions and to improve the manuscript.
Regarding the issue about that "Reference conditions were selected based on natural land use in the catchment, lack or low human activity, and absence of point sources of pollution as well as good water quality" (lines #11and #12), these criteria cannot be used to select reference conditions, but as a previous filter. As explained in my previous revision, it is obvious that lakes used to derive reference conditions should be as pristine as possible.
In the same way, I agree with the authors in their explanation about the methods to be used in establishing reference conditions (cover letter), whenever "anthropogenic impacts" are not considered among environmental variables for that purpose. As explained also in my previous revision, it is obvious that lakes used to derive reference conditions should be as pristine as possible. Among variables used by the authors, only "trophic level" could be then used for that purpose, provided that it does not significantly result from anthropogenic impacts, but from natural processes.
Finally, talking about reference conditions for stratified lakes requires, at least, more details than just saying that lakes are stratified. I regret to say that it is not acceptable to cope with this task without even saying whether lakes are monomictic, dimictic, polymictic, etc. The authors have not addressed this issue, as recommended in my previoous revision.
I regret to say that I do not feel comfortable in recommending the manuscript for publication unless the authors cope with these problems.

Round 3
Reviewer 2 Report
Thank you very much to the authors for providing the information about startificaton that helps to establish the reference conditions for the studied lakes.
Regarding the concept of reference conditions, I humbly urge the authors to stick to evidences and scientific work, not to "point of view" or "philosophy".
I agree in arguments given by the author in the late cover letter, as in previous arguments given in previous cover letter about this question, but the problem is that the authors do not apply this concept of reference conditions (this has been extensively explained in my previous reviews).
To help the authors to solve this question, I made a literature review and found a publication that they might have used as a reference, and that I encourage them to use it now. I talk about https://doi.org/10.1016/j.limno.2005.04.001, which has been cited 76 times in other scientific publications and deals with Polish lakes.
That work is consistent with my previous reviews and establishes a start point for this valuable work of the authors.
The conceptual diagram for "state of the art" that authors might be willing to follow could be as follows: 1) The above mentioned work by Kolada et al. (2005), 2) The work by Jarvinen et al. (2013) on reference conditions usinf phytoplankton, and 3) Author's own work.
As explained in my previous reviews, authors might start filtering lakes that can be used to establish reference conditions (i.e., those that according to the variables measuring anthropogenic influence are acceptably conserved), and proceed then to follow, for instance, the suggested procedure, or any other THAT DO NOT EQUALS REFERENCE CONDITIONS TO CONSERVATION STATUS.
Author Response
Thank you very much for your time to prepare the next review. We hope that we addressed your comment accordingly and resolved all your concerns, and that you ultimately find our revision satisfactorily.
We hope that the recommended literature position (https://doi.org/10.1016/j.limno.2005.04.001) allowed us to understand the Reviewer's way of thinking about the method of elaboration reference conditions. Our mistake was not to provide information about the abiotic typology of the studied lakes in the text. Selection of lakes to investigation was based on new abiotic typology according to Kolada et al. (2017), which is the updated version of the typology from 2005 (Kolada et al. 2005). Information about studied lakes’ typology has been added in lines 75-78, „The lakes were selected to represent homogenous abiotic conditions, i.e., lowland, highly alkaline, stratified ecosystems on calcareous deposits [26] as well as the entire gradient of ecological conditions, in terms of trophic level and anthropogenic impacts”.
References:
Kolada, A.; Soszka, H.; Kutyła, S.; Pasztaleniec, A. The typology of Polish lakes after a decade of its use: A critical review and verification. 2017. Limnologica, 67, 20-26.
Round 4
Reviewer 2 Report
Thank you very much for your effort. You must be proud of the results of your revising task. The manuscript has been now significantly improved.
There is just one minor issue left about the "reference conditions" issue. Please, modify lines #11 and #12 in 'Abstract' accordingly.
Author Response
Thank you very much for the effort you put into the review of our paper.
- There is just one minor issue left about the "reference conditions" issue. Please, modify lines #11 and #12 in 'Abstract' accordingly. Information about abiotic conditions of studied lakes has been added in Abstract.